# Performance Evaluation of RNN with Hyperbolic Secant in Gate Structure through Application of Parkinson's Disease Detection

**Tomohiro Fujita, Zhiwei Luo, Changqin Quan \*, Kohei Mori and Sheng Cao**

Graduate School of System Informatics, Kobe University, 1-1, Rokkodai-cho, Nada-ku, Kobe 657-8501, Japan; tomohiro@cs11.cs.kobe-u.ac.jp (T.F.); luo@gold.kobe-u.ac.jp (Z.L.); mori@cs.kobe-u.ac.jp (K.M.); jasonsosen@gold.kobe-u.ac.jp (S.C.)
* **Correspondence: quanchqin@gold.kobe-u.ac.jp**

**Abstract:** This paper studies a novel recurrent neural network (RNN) with hyperbolic secant (sech) in the gate for a specific medical application task of Parkinson's disease (PD) detection. In detail, it focuses on the fact that patients with PD have motor speech disorders, by converting the voice data into black-and-white images of a recurrence plot (RP) at specific time intervals and constructing the detection model that combines RNN and convolutional neural network (CNN); the study evaluates the performance of the RNN with sech gate compared with long short-term memory (LSTM) and gated recurrent unit (GRU) with conventional gates. As a result, the proposed model obtained similar results to LSTM and GRU (an average accuracy of about 70%) with less hyperparameters, resulting in faster learning. In addition, in the framework of the RNN with sech in gate, the accuracy obtained by using tanh as the output activation function is higher than using the relu function. The proposed method will see more improvement by increasing the data in the future. More analysis on the input sound type, the RP image size, and the deep learning structures will be included in our future work for further improving the performance of PD detection from voice.

**Keywords:** recurrent neural network (RNN); hyperbolic secant (sech) function; recurrence plot; Parkinson's disease

## 1. Introduction

In recent years, the recurrent neural network (RNN) has been frequently used in time series data processing such as in medical information processing, etc. The RNN has a recursive structure inside and can handle variable lengths of input data.

Generally, since it is difficult for a Simple RNN (Vanilla RNN) [1] with a simple structure to learn the time series data with long-term dependencies, two types of RNNs with complex gated structures to control the required information are proposed; they are long short-term memory (LSTM) [2,3] and gated recurrent unit (GRU) [4], respectively. However, while the performance of RNNs with gated structures is improved, since back-propagation through time (BPTT) used for learning works by unrolling all input time steps, the more parameters there are in the RNN, the more memory is required and the higher the calculation costs. In order to solve this problem, many papers have been published that attempted to simplify LSTM and GRU [5–8]. In our previous research, we proposed SGR (simple gated RNN) that uses the parts of gate structure of LSTM and GRU for Simple RNN to reduce parameters and realize faster learning [9]. In addition, we also proposed an RNN that introduced a new activation function "hyperbolic secant (sech)" into the gate [10], which is even simpler than Simple RNN (Vanilla RNN).

The objective of this paper is to evaluate this RNN model with sech in the gate through a specific medical application task. In detail, the task here is to detect Parkinson's disease (PD) from the sound information of subjects. PD is the second most common

neurodegenerative disease after Alzheimer's disease [11]. The detection of PD is very important for medical treatments as well as improving the patient's quality of live (QOL). Focusing on the fact that patients with PD have motor speech disorders, here we ask subjects to pronounce a voice of /a/ for about 4 to 10 s to acquire data. In order to check the periodicity of the voice, the sound data are then converted into a recurrence plot (RP) at specific time intervals. A recurrence plot (RP) can visualize a periodic nature and chaos in time series. The generated RPs are set as input to a neural network model that combines the convolutional neural network (CNN) and RNN for classification.

In the experiments, in order to evaluate the performance of the RNN with sech gate structure, we compared it with LSTM and GRU with conventional gate structures. We also compared the performance of this RNN when the used output activation function was changed between tanh and relu. For performance evaluation, accuracy, F1-score and Matthews correlation coefficient (MCC) were used. As a result, the proposed model obtained similar results to LSTM and GRU (an average accuracy of about 70%) with less hyperparameters, contributing to faster learning. In addition, in the framework of the RNN with sech in gate, the accuracy obtained by using tanh as the output activation function is higher than using the relu function.

The outline of the paper is as follows: Section 2 discusses related works. Section 3 explains the framework of PD detection models and data preprocessing. The details of the experiments, the results, and discussion are described in Section 4. Section 5 is the conclusions.

## 2. Related Work

In this section, we review important points regarding related works on RNNs and CNNs and the recurrence plot used in our study.

As mentioned in Section 1, recent RNNs generally refer to RNNs with weighted gate structures such as LSTM and GRU, rather than Simple RNN (or Vanilla RNN) with simple structures. This is because the RNN with gate structure succeeded in partially solving a vanishing gradient problem. When the vanishing gradient occurs, the error becomes smaller rapidly as it goes back layers, and the learning does not progress well. The gate structure deals with this problem by controlling the vanishing gradient with weight parameters. However, the weight parameters increase the calculation cost, and the analysis is difficult. To address these problems, many papers have been written that attempted to reduce the parameters of the RNN [5–8]. As in our previous study, we proposed SGR that reduced weighted gates while maintaining the performance. However, due to having weighted gates, the calculation costs were not significantly reduced, and the analysis remained difficult [9]. Therefore, we proposed a new RNN model that reduced parameters by removing the conventional weighted gate and using a new gate structure with scalar value controlling the vanishing gradient and the sech function as the activation function [10]. Regarding this RNN using the sech function, in [10], the task performance was lower than that of conventional gated RNNs (such as LSTM and GRU) due to the characteristics of the specific task and tanh activation function. However, in a binary classification task in natural language processing (NLP), it was found that, despite the parameter of about 1/6 or less, the performance was comparable to that of conventional gated RNNs without using normalization methods such as batch normalization. It is then important to clarify the performance quantity for more tasks such as time series data processing and image processing.

In this paper, we confirm the performance difference in a practical medical application task between this RNN with sech function and conventional gated RNNs and evaluate it. Therefore, in this study, we set the detection of Parkinson's disease as the practical medical application task. In time-frequency analysis for analyzing time series data, it is common to use the short-time Fourier transform (STFT). STFT divides a time signal into short segments of equal length and computes the Fourier transform separately on each segment. However, Fourier transform has the drawback of lower resolution for non-stationary signals. Since the voice of PD patients is non-stationary due to a faint or unstable

voice caused by dysarthria, in our study, we attempt voice analysis using a recurrence plot (RP) that can process even non-stationary signals. As the way to represent chaotic time series and visualize a periodic nature, a recurrence plot (RP) was introduced by Eckmann et al. [12]. There are various methods for generating recurrence plots, such as plotting points that are smaller than the arbitrary threshold or generating recurrence plots using percentile. Compared to the Fourier transform, which is not suitable for describing systems in which independent basis functions cannot be properly selected, RP can handle both non-linear and unsteady states [13]. Additionally, RP can detect even faint modulations of animal voice signals that cannot be captured by conventional time-frequency analysis and is a very powerful tool [14].

Recently, there has been increasing research in which recurrence plot images are classified using a CNN [15,16], which is used for the image recognition tasks [17–19]. Furthermore, there is a Parkinson's disease identification study which used a CNN and recurrence plots of handwriting dynamics data, too [20]. Therefore, in this paper, we propose a Parkinson's disease voice detection model that combines a CNN and RNN and use it to evaluate RNNs.

Related to other studies on PD detection, in the machine learning approach, there is a study that used embedding extracted from a deep neural network named x vectors and classified it using cosine distance, cosine distance preceded by Latent Dirichlet Allocation (LDA), and Polylingual Latent Dirichlet Allocation (PLDA) [21]. Additionally, in [22], four machine learning methods (k-nearest neighbor, multi-layer perceptron, optimum-path forest, and support vector machine) were used with 18 feature extraction techniques for the detection of PD. In the deep learning approach, in [23], multiple artificial neural networks (ANNs) were used with 26 speech features for PD detection, and principal component analysis (PCA) and self-organizing map (SOM) were applied for feature selection. While in [24], a deep neural network (DNN) was applied for PD severity prediction using 16 biomedical voice measures. In this research, the calculation costs of voice preprocessing are relatively high, and multiple voice features are required. Compared with this research, in our study, we propose to use an RP which only calculates the percentile and absolute distance so as to reduce the calculation cost and make it easy for implementation. By using an RP generated from simple vowel voice, our approach uniquely detects PD with the model combining the CNN and RNN. To the best of our knowledge, detecting PD with the model combining the CNN and RNN using an RP generated from simple vowel voice has not been referred before.

## 3. The Model Description and Data Preprocessing

This section describes neural network models used in this paper in detail.

### 3.1. RNN Models

3.1.1. Long Short-Term Memory (LSTM)

As is shown in Figure 1, a layer of typical LSTM is defined by Equations (1)–(6) as follows:

$$z_t = \tanh(W_z x_t + U_z h_{t-1} + b_z) \tag{1}$$

$$i_t = \text{sigmoid}(W_{in} x_t + U_{in} h_{t-1} + b_{in}) \tag{2}$$

$$o_t = \text{sigmoid}(W_{out} x_t + U_{out} h_{t-1} + b_{out}) \tag{3}$$

$$f_t = \text{sigmoid}\left(W_{for} x_t + U_{for} h_{t-1} + b_{for}\right) \tag{4}$$

$$c_t = i_t \circ z_t + f_t \circ c_{t-1} \tag{5}$$

$$h_t = \tanh(c_t) \circ o_t \tag{6}$$

where $z_t$ is an input to be added to cell state, $c_t$ is an internal cell state, $h_t$ is a hidden state at next time. Additionally, $i_t$, $o_t$, $f_t$ represent outputs from input gate, output gate, and forget gate, respectively. From here, unless otherwise specified, the operator symbol $\circ$ represents

the Hadamard product, $W$, $U \in \mathbb{R}^{m \times n}$ are weighting matrices, $b \in \mathbb{R}^n$ is bias, regardless of the subscript, and the explanations are omitted hereafter. In addition, $x_t$ is an input at time step $t$, $h_{t-1}$ is a hidden state at time step $t - 1$, and $h_t$ is a hidden state at time step $t$ in this paper.

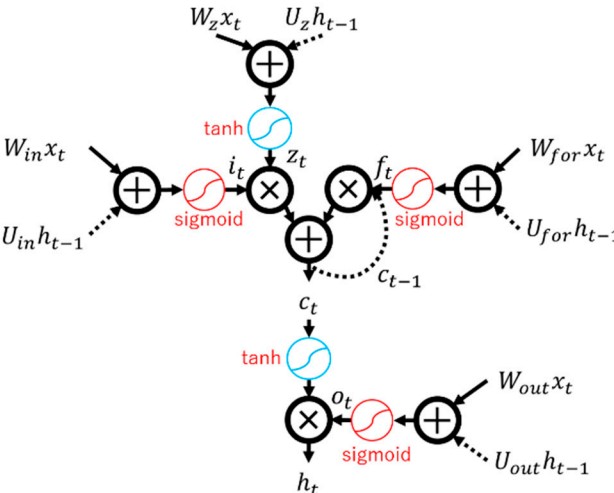

**Figure 1.** A layer of typical LSTM. LSTM has three gates with weight parameters to control the information that should be retained. The biases are omitted here.

The input gate controls input to cell state, using the value of $i_t$ $(0 < i_t < 1)$ which is the output from Equation (2) so that it can be selectively stored from the input. Additionally, the output gate controls the output from cell state using the value of $o_t$ $(0 < o_t < 1)$ which is the output from Equation (3). Finally, the forget gate controls the internal cell state directly by $f_t$ $(0 < f_t < 1)$ which is the output from Equation (4) [2,3].

### 3.1.2. Gated Recurrent Unit (GRU)

As is shown in Figure 2, a layer of GRU is defined by Equations (7)–(10) as follows:

$$r_t = \text{sigmoid}(W_r x_t + U_r h_{t-1} + b_r) \tag{7}$$

$$u_t = \text{sigmoid}(W_u x_t + U_u h_{t-1} + b_u) \tag{8}$$

$$\widetilde{h_t} = \tanh\left(W_{gr} x_t + U_{gr}(r_t \circ h_{t-1}) + b_{gr}\right) \tag{9}$$

$$h_t = u_t \circ h_{t-1} + (1 - u_t) \circ \widetilde{h_t} \tag{10}$$

where $r_t$, $u_t$ are outputs from reset gate and update gate. The reset gate controls how much the previous hidden state $h_{t-1}$ is considered to create a new hidden state $\widetilde{h_t}$, using the gate output $r_t$ $(0 < r_t < 1)$ which is the output from Equation (7). Similarly, the update gate is in control of deciding how much the new hidden state $\widetilde{h_t}$ is mixed to generate the next hidden state $h_t$, using the gate output $u_t$ $(0 < u_t < 1)$ which is the output from Equation (8) [4]. The performance of GRU compared to that of LSTM depends on the learning task, while the required number of weight parameters for the same number of units is 3/4 that of LSTM.

### 3.1.3. Our Proposed RNN Model

In the conventional method, there are various problems such as an increase in the amount of calculation, difficulty in the analysis of the gate structure, and dependence on data due to batch normalization. Therefore, in order to solve these problems in the conventional RNNs, we constructed the new structure using the sech function as the gate structure [10]. In the conventional gate structure, if the gate structure has weight parameters, the calculation cost is very high, but if it does not have weight parameters, scaling for negative and positive values does not work properly, because the sigmoid

function is not an even function. Hence, instead of the sigmoid function, we used the sech function that is an even function and has an output range from 0 to 1 for the gate structure [10].

The sech function is defined by Equation (11) as follows:

$$\text{sech}(x) = 2/\left(e^x + e^{-x}\right) \tag{11}$$

Additionally, differentiating (11) is defined by Equation (12) as follows

$$\frac{d}{dx}\left(\text{sech}(x)\right) = -\text{sech}(x)\cdot\text{tanh}(x) \tag{12}$$

Figure 3 shows a graph of the sech function and sech's differentiated function.

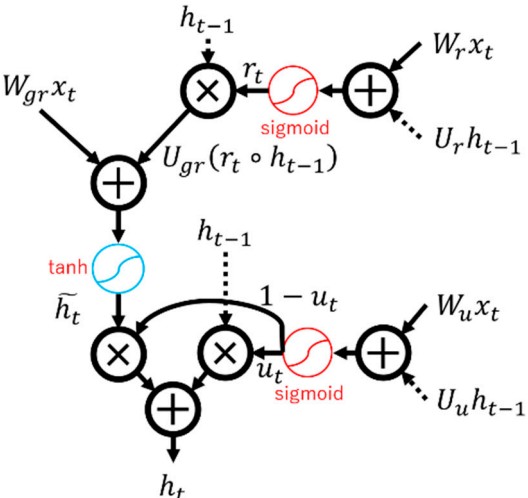

**Figure 2.** A layer of GRU. GRU controls how much information is retained by two gates, reset gate and update gate. The biases are omitted here.

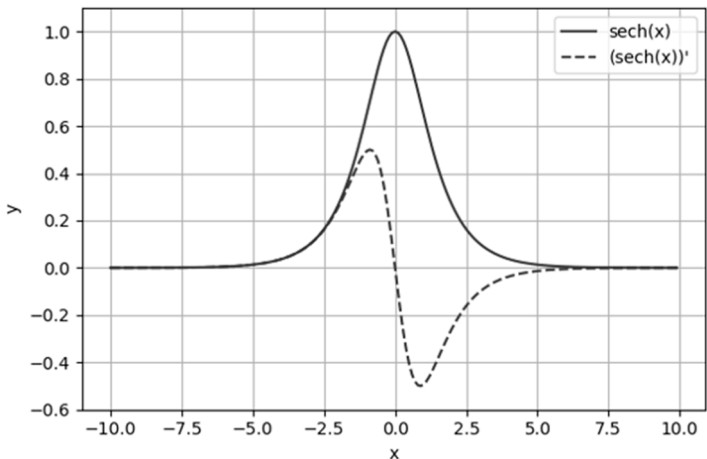

**Figure 3.** A graph of the sech function and sech's differentiated function (cited from [10]). Sech function is even function and has output range from 0 to 1.

As is shown in Figure 4, a layer of the RNN with sech gate structure with the highest performance in the paper [10] is defined by Equations (13)–(15) as follows:

$$\widetilde{h}_t = \text{sech}(ah_{t-1}) \circ h_{t-1} \tag{13}$$

$$h_t = (W_x x_t + b_x) + \widetilde{h}_t \tag{14}$$

$$o_t = \text{act}(h_t) \tag{15}$$

where $a$ is a scalar value, which is introduced as a parameter for controlling the degree of vanishing gradient. Additionally, the output to the lower layer is represented by $o_t$, and act is activation function. Hereinafter, for the sake of simplicity, the RNN with sech gate structure is referred to as "RNN-SH".

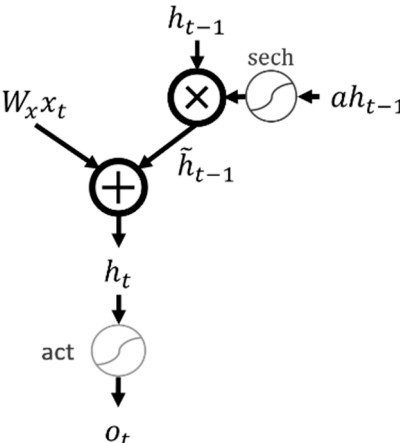

**Figure 4.** A layer of RNN-SH (cited from [10]). The smaller scalar value $a$ is, the larger the hidden state and the gradient are after passing through the gate, and the larger $a$ is, the smaller they are. This $a$ controls the amount of retained information and the degree of vanishing gradient. The biases are omitted here.

### 3.2. Parkinson's Disease Detection

### 3.2.1. Voice Data Preprocessing and Recurrence Plot Creation

Here, we explain how to preprocess voice data and create recurrence plots. The voice data used in our experiments are recordings of the voice/a/ including men and women for about 4 to 10 s. Regardless of the recording, all voice data are converted to monaural, 16,000 Hz, and then, processing to create RPs is started, because voice data are recorded in various environments. Additionally, all voice data are quantized in 16 bits.

The preprocessing of voice data before creating RPs is as follows:

1.  Delete the silent section before and after voice data: In order to avoid the influence of voice volume, all voice data are normalized so that the maximum amplitude is the expressible maximum value, and the sections of $-40$ dB or less are deleted only from before and after the sound.
2.  Furthermore, delete the first and last of the above voice data for 0.1 s: This was because the first and last sounds after deleting the silent sections were often unstable.

Next, the procedure of creating recurrence plots is as follows:

1.  Divide the above preprocessed voice data into 0.01 s sections.
2.  Immediately before creating the recurrence plot, normalize the sound of 0.01 s section so that it becomes $[-1, 1]$, and then plot points with a distance smaller than the 35th percentile in 0.01 s section. The reason for using the 35th percentile is that the accuracy was highest when the 35th percentile was used in the experiments, which is described later.
3.  Compress the generated black-and-white RP with $160 \times 160$ image size to $20 \times 20$ using bilinear interpolation. This may deteriorate the accuracy, but it is carried out in consideration of memory efficiency of Video Random Access Memory (VRAM) when inputting to the neural network.
4.  Repeat above 1–3 until all divided voice data become recurrence plots. However, if a fraction less than 0.01 s appears in the last section, it will be rounded down.

### 3.2.2. Parkinson's Disease Detection Model

Figure 5 shows the structure of the Parkinson's disease detection model. The CNN model, RNN model, and output layer are processed in this order for detection. Relu is used as the activation function in the CNN model, but for each RNN, a suitable activation function is used (tanh or relu).

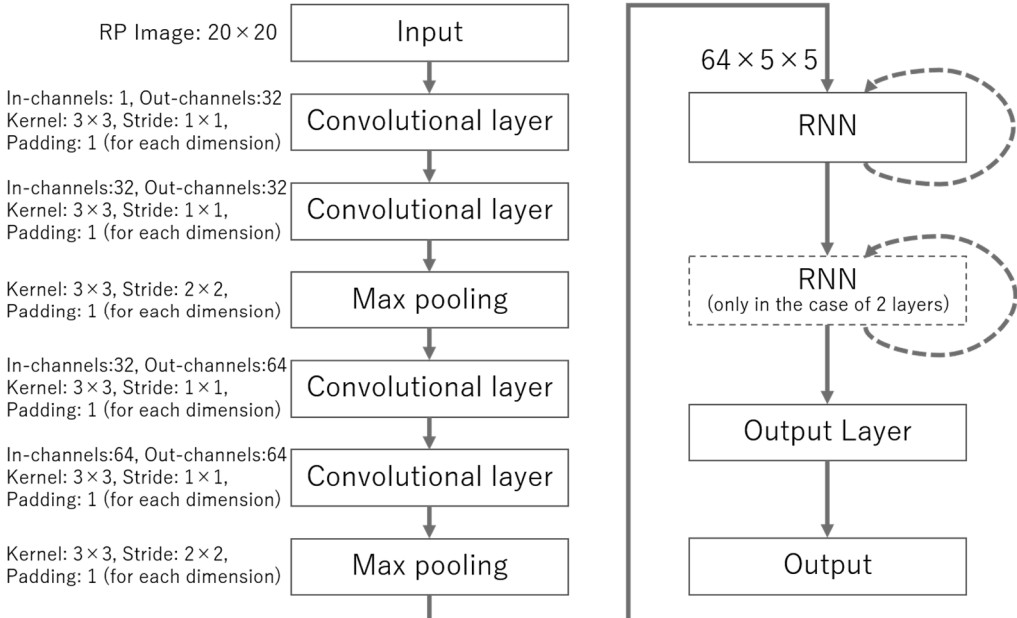

**Figure 5.** The structure of Parkinson's disease detection model. The input image size (RP) is $20 \times 20$, and the relu was used as the activation function after each output of the convolutional layer. In order to prevent overfitting, dropout is applied immediately after max pooling and before output layer. If the RNN has two layers, dropout is applied before the RNN of the second layer too. The size input to RNN is $64 \times 5 \times 5$ (= 1600) because the output from the CNN is flattened. The model output is binary (2 classes), and the output is generated after all RPs have been input.

## 4. Performance Evaluation

### 4.1. Outline of Experiments

In order to evaluate the performance of each RNN using the recurrence plots and the model constructed in Section 3, experiments were performed using a computer. Here, PyTorch 1.7.1 which is a machine learning library, cuda11 and Python 3 (version 3.8) was used. The execution environment was Ubuntu 18.04.4 (64 bit) and we used i7 8700 (RAM 16 GB) and gtx1080 of GPU (VRAM 8 GB) for single precision calculations. For the processing of voice data, a python library named Pydub [25] was used. Some of Pydub functions depend on FFmpeg [26]. In addition, when using GPU on PyTorch, it was necessary to set CUBLAS_WORKSPACE_CONFIG =: 16:8 as an environment variable for reproducibility due to use of cuda11. In the experiments, all random seeds were initialized to 10.

### 4.2. Input Voices and Preprocessing

As mentioned in Section 3, the voice dataset used in the experiments are recordings of the voice /a/ for about 4 to 10 s. This dataset contains data for 22 healthy people (HP) and 30 PD patients, and three times data for each person is recorded. Therefore, the total number of data is 156 ((22 + 30) × 3). The dataset contains 43 subjects (13 HP and 30 PD cases) who were hired as volunteers by the GYENNO SCIENCE Parkinson Disease Research Center (Ethical Approval: All procedures performed in studies involving human participants were in accordance with the ethical standards of the institutional and/or national research committee and the "Law of the People's Republic of China on Medical Practitioners" (1998)

declaration and its later amendments or comparable ethical standards), and nine subjects (nine HP) who were requested by the authors in order to alleviate the data imbalance. The breakdown of gender was 27 females and 25 males. PD patients consisted of patients with HY (Hoehn and Yahr) stage 1–5. A total of 13 PD patients were in stage 3–5, and 17 PD patients were lower than stage 3.

Figure 6 shows the examples of preprocessed voice data and generated RPs in the procedure of Section 3. If the value of the 35th percentile is $p$ for the time series $\{x_i\}_{i=1}^{N} = \{x_1, x_2, \cdots, x_N\}$, the recurrence plot matrix $R$ is plotted by Equation (16) as follows:

$$R(i, j) = \begin{cases} 1 & \text{if } |x_i - x_j| < p \\ 0 & \text{otherwise,} \end{cases} \tag{16}$$

where $i, j \in \{1, 2, \cdots, N\}$ are the numbers of the components. $R(i, j) = 1$ means that black is plotted, and $R(i, j) = 0$ means that white is plotted.

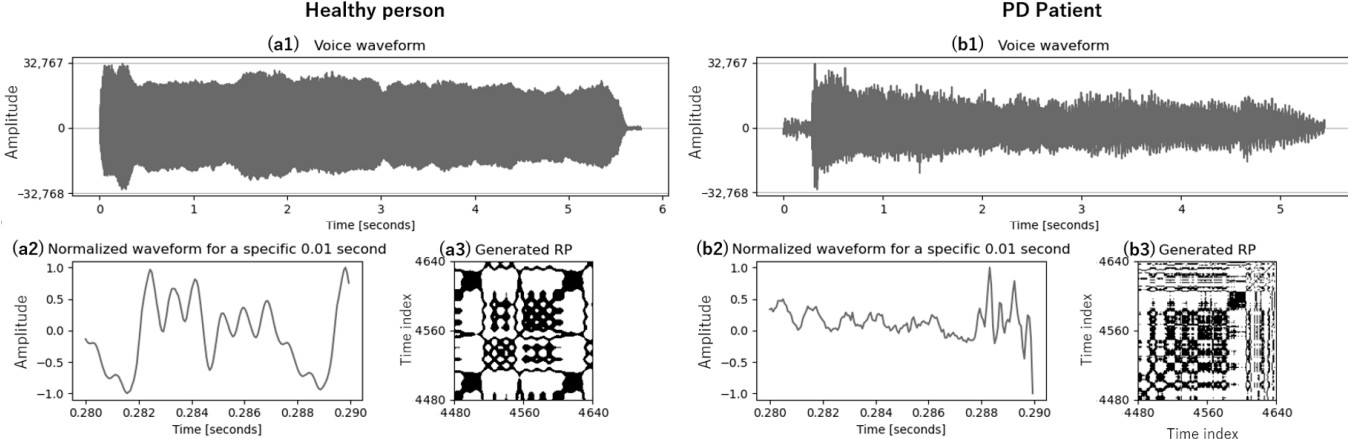

**Figure 6.** The examples of preprocessed voice data and generated RPs. The left image (**a**) is an example of a healthy person, and the right image (**b**) is an example of a PD patient. (**a1**,**b1**) are voice waveform, (**a2**,**b2**) are normalized waveform for 0.280 to 0.290 s. (**a3**,**b3**) are generated RPs from normalized waveform for 0.280 to 0.290 s. PD patients have a faint voice by dysarthria and there are differences between the two recurrence plots. The RP was rotated 90 degrees in this figure.

We divided dataset into, train: $28 \times 3$ (HP: $10 \times 3$, PD: $18 \times 3$, about 54%), validation: $12 \times 3$ (HP: $6 \times 3$, PD: $6 \times 3$, about 23%), and test: $12 \times 3$ (HP: $6 \times 3$, PD: $6 \times 3$, about 23%). However, the three data of the same person were divided so that they belonged to the same group (train or validation or test) in the experiments. Additionally, at the time of training, the voices of healthy person data shifted by 2 s were used as oversampling data so that the total number of data did not become imbalanced. Furthermore, in order to avoid the influence of dispersion due to data division, the data were shuffled, experiments were performed five times consecutively for each RNN, and the performance was evaluated based on the average.

### 4.3. RNN Model Configuration and Hyperparameters

The RNNs used in the experiments were LSTM, GRU, and RNN-SH. Additionally, the number of hidden units was set to 64, 128, and 256. RNN-SH used tanh or relu as the output activation function. We also experimented when the RNNs are stacked in two layers. We used RAdam [27], which is a kind of stochastic gradient descent algorithm for learning, and set RAdam parameters to recommended values in Adam [28]. The dropout ratio and parameter of weight decay were set to 0.5 and 0.01, respectively, to prevent overfitting. The number of epochs was 50, and the mini-batch size was 27. Regarding the initialization of weight parameters, they were initialized to a Gaussian distribution with a mean of 0 and a variance of $1/n$ ($n$: the number of input units) [29], and all biases were initialized to 0. Additionally, CNN weights were initialized to a uniform distribution $[-v, v]$ where

$v = \sqrt{3/n}$ [29]. The scalar value $a$ of gate structure of RNN-SH was initialized with 1.0. Finally, Softmax cross entropy was used as the loss function. This loss function ($L$) is expressed by the following Equation (17):

$$L = -\frac{1}{m} \sum_{i,j} t_{ij} \log\left( \frac{\exp(y_{ij})}{\sum_{k=1}^{o} \exp(y_{ik})} \right) \tag{17}$$

where $m$ is the batch size, $o$ is the output size, $y_{ij}$ is the output from output layer, and $t_{ij}$ is the correct answer label. In addition, $i$ is the data number, $j$ is the dimension, and $t_{ij}$ is the correct answer label of $j$-th dimension in the $i$-th data.

### 4.4. Experimental Results

The results of the experiments are shown in Table 1. In addition, Figure 7 shows graphs of the average values in Table 1 of each model. Table 1 shows the results of the validation set and test set at the time when the loss was the lowest with respect to the validation set in 50 epochs. Since the experiments were performed five times each, the mean and standard deviation of the validation set and the test set are respectively shown. Additionally, the overall mean and standard deviation of the validation set and test set are shown too. "Average (validation/test)" is the average of five times for each validation set or each test set, respectively. "Total Average" is the all average of the validation set and test set. Here, we used accuracy, *F-score* and Matthews correlation coefficient (*MCC*) as performance evaluation indicators. Each formula is defined by Equations (18)–(20) as follows:

$$Accuracy = (TP + TN)/(TP + TN + FP + FN) \tag{18}$$

$$F - score = 2TP/(2TP + FP + FN) \tag{19}$$

$$MCC = \frac{(TP \times TN) - (FP \times FN)}{\sqrt{(TP + FP) \times (TP + FN) \times (TN + FP) \times (TN + FN)}}, \tag{20}$$

where $TP$, $TN$, $FP$, $FN$ are the number of true positives, true negatives, false positives, and false negatives, respectively. Accuracy and *F-score* is a value between 0 and 1, the closer they are to 1, the better the performance is. For unbalanced data, *F-score* can perform more accurate performance evaluation than accuracy. Note that the *F1-score* shown in the experiments is macro-average. The *MCC* is a correlation coefficient value between $-1$ and $+1$, and $+1$ indicates a perfect prediction, 0 an average random prediction and $-1$ an inverse prediction. The *MCC* was calculated for more accurate evaluation, including inverse predictions. In Table 1, accuracy and F-score show the value rounded off the third decimal place, and *MCC* shows the value rounded off the fourth decimal place.

**Table 1.** Experimental results (validation set/test set).

| Layers | Model | Units | Indicators | The Number of Trials 1 | 2 | 3 | 4 | 5 | Average (Validation/Test) | Total Average |
|---|---|---|---|---|---|---|---|---|---|---|
| 1 | LSTM | 64 | Acc. (%) | 75.00/72.22 | 77.78/55.56 | 58.33/75.00 | 69.44/63.89 | 61.11/63.89 | 68.33 ± 7.58/66.11 ± 6.89 | 67.22 ± 7.33 |
| | | | F1. (%) | 74.98/72.14 | 77.50/55.42 | 58.04/74.51 | 69.42/63.18 | 60.63/62.47 | 68.11 ± 7.67/65.54 ± 6.95 | 66.83 ± 7.43 |
| | | | MCC | 0.501/0.447 | 0.570/0.112 | 0.169/0.521 | 0.390/0.289 | 0.228/0.302 | 0.371 ± 0.154/0.334 ± 0.141 | 0.353 ± 0.149 |
| | | 128 | Acc. (%) | 69.44/69.44 | 75.00/55.56 | 66.67/69.44 | 61.11/72.22 | 44.44/27.78 | 63.33 ± 10.45/58.89 ± 16.61 | 61.11 ± 14.05 |
| | | | F1. (%) | 69.23/68.84 | 74.02/55.42 | 66.56/68.24 | 60.63/72.22 | 44.44/27.55 | 62.98 ± 10.23/58.46 ± 16.48 | 60.72 ± 13.90 |
| | | | MCC | 0.394/0.405 | 0.543/0.112 | 0.335/0.422 | 0.228/0.444 | -0.111/-0.447 | 0.278 ± 0.219/0.187 ± 0.340 | 0.233 ± 0.290 |
| | | 256 | Acc. (%) | 75.00/77.78 | 80.56/61.11 | 72.22/61.11 | 61.11/72.22 | 55.56/77.78 | 68.89 ± 9.20/70.00 ± 7.54 | 69.44 ± 8.43 |
| | | | F1. (%) | 74.98/77.71 | 80.17/60.99 | 70.78/59.09 | 60.63/72.14 | 55.42/77.78 | 68.40 ± 9.13/69.54 ± 8.04 | 68.97 ± 8.62 |
| | | | MCC | 0.501/0.559 | 0.636/0.224 | 0.496/0.248 | 0.228/0.447 | 0.112/0.556 | 0.395 ± 0.194/0.407 ± 0.145 | 0.401 ± 0.171 |
| | GRU | 64 | Acc. (%) | 69.44/75.00 | 80.56/58.33 | 63.89/63.89 | 63.89/55.56 | 58.33/77.78 | 67.22 ± 7.54/66.11 ± 8.85 | 66.67 ± 8.24 |
| | | | F1. (%) | 69.42/74.98 | 80.17/58.30 | 63.18/60.17 | 63.86/55.42 | 58.04/77.78 | 66.94 ± 7.54/65.33 ± 9.19 | 66.13 ± 8.44 |
| | | | MCC | 0.390/0.501 | 0.636/0.167 | 0.289/0.351 | 0.278/0.112 | 0.169/0.556 | 0.352 ± 0.158/0.337 ± 0.176 | 0.345 ± 0.167 |
| | | 128 | Acc. (%) | 69.44/77.78 | 75.00/63.89 | 69.44/66.67 | 66.67/72.22 | 47.22/52.78 | 65.56 ± 9.56/66.67 ± 8.43 | 66.11 ± 9.03 |
| | | | F1. (%) | 69.42/77.78 | 73.33/62.47 | 69.42/62.50 | 66.56/72.22 | 46.18/49.63 | 64.98 ± 9.64/64.92 ± 9.64 | 64.95 ± 9.64 |
| | | | MCC | 0.390/0.556 | 0.577/0.302 | 0.390/0.447 | 0.335/0.444 | -0.058/0.064 | 0.327 ± 0.209/0.363 ± 0.170 | 0.345 ± 0.191 |
| | | 256 | Acc. (%) | 69.44/75.00 | 83.33/58.33 | 75.00/61.11 | 72.22/69.44 | 52.78/69.44 | 70.56 ± 10.03/66.67 ± 6.09 | 68.61 ± 8.52 |
| | | | F1. (%) | 69.42/74.98 | 83.13/58.30 | 74.83/57.86 | 72.22/68.84 | 51.85/69.23 | 70.29 ± 10.29/65.84 ± 6.70 | 68.07 ± 8.97 |
| | | | MCC | 0.390/0.501 | 0.684/0.167 | 0.507/0.267 | 0.444/0.405 | 0.058/0.394 | 0.417 ± 0.205/0.347 ± 0.117 | 0.382 ± 0.170 |
| | RNN-SH (tanh) | 64 | Acc. (%) | 72.22/69.44 | 83.33/58.33 | 63.89/69.44 | 75.00/63.89 | 61.11/55.56 | 71.11 ± 7.97/63.33 ± 5.67 | 67.22 ± 7.93 |
| | | | F1. (%) | 72.14/68.84 | 83.13/57.51 | 61.48/69.23 | 74.98/63.18 | 59.09/50.00 | 70.16 ± 8.87/61.75 ± 7.27 | 65.96 ± 9.13 |
| | | | MCC | 0.447/0.405 | 0.684/0.174 | 0.321/0.394 | 0.501/0.289 | 0.248/0.149 | 0.440 ± 0.151/0.282 ± 0.107 | 0.361 ± 0.153 |
| | | 128 | Acc. (%) | 75.00/72.22 | 77.78/63.89 | 66.67/63.89 | 69.44/61.11 | 55.56/55.56 | 68.89 ± 7.74/63.33 ± 5.39 | 66.11 ± 7.22 |
| | | | F1. (%) | 74.98/71.88 | 77.14/63.64 | 65.71/63.18 | 69.42/60.63 | 47.64/55.00 | 66.98 ± 10.48/62.86 ± 5.45 | 64.92 ± 8.60 |
| | | | MCC | 0.501/0.456 | 0.589/0.282 | 0.354/0.289 | 0.390/0.228 | 0.177/0.114 | 0.402 ± 0.140/0.274 ± 0.111 | 0.338 ± 0.142 |
| | | 256 | Acc. (%) | 72.22/80.56 | 83.33/61.11 | 72.22/66.67 | 63.89/72.22 | 61.11/69.44 | 70.56 ± 7.78/70.00 ± 6.43 | 70.28 ± 7.14 |
| | | | F1. (%) | 72.22/80.54 | 83.13/61.11 | 72.14/63.88 | 63.64/72.22 | 59.09/68.84 | 70.04 ± 8.26/69.32 ± 6.80 | 69.68 ± 7.58 |
| | | | MCC | 0.444/0.612 | 0.684/0.222 | 0.447/0.401 | 0.282/0.444 | 0.248/0.405 | 0.421 ± 0.155/0.417 ± 0.124 | 0.419 ± 0.140 |
| | RNN-SH (relu) | 64 | Acc. (%) | 63.89/77.78 | 44.44/50.00 | 63.89/69.44 | 58.33/66.67 | 55.56/75.00 | 57.22 ± 7.16/67.78 ± 9.72 | 62.50 ± 10.04 |
| | | | F1. (%) | 63.64/77.71 | 37.50/37.69 | 58.47/69.23 | 57.51/66.56 | 55.00/74.98 | 54.42 ± 8.92/65.24 ± 14.33 | 59.83 ± 13.10 |
| | | | MCC | 0.282/0.559 | -0.149/0.000 | 0.402/0.394 | 0.174/0.335 | 0.114/0.501 | 0.164 ± 0.185/0.358 ± 0.195 | 0.261 ± 0.213 |
| | | 128 | Acc. (%) | 75.00/66.67 | 66.67/63.89 | 66.67/66.67 | 63.89/69.44 | 55.56/52.78 | 65.56 ± 6.24/63.89 ± 5.83 | 64.72 ± 6.09 |
| | | | F1. (%) | 74.83/65.71 | 62.50/60.17 | 64.94/66.25 | 63.64/69.42 | 44.62/39.23 | 62.10 ± 9.78/60.16 ± 10.88 | 61.13 ± 10.39 |
| | | | MCC | 0.507/0.354 | 0.447/0.351 | 0.372/0.342 | 0.282/0.390 | 0.243/0.169 | 0.370 ± 0.099/0.321 ± 0.078 | 0.346 ± 0.092 |
| | | 256 | Acc. (%) | 66.67/75.00 | 83.33/61.11 | 69.44/63.89 | 66.67/75.00 | 63.89/44.44 | 70.00 ± 6.89/63.89 ± 11.25 | 66.94 ± 9.82 |
| | | | F1. (%) | 66.25/74.83 | 83.13/61.11 | 68.24/63.64 | 66.56/74.98 | 63.64/42.86 | 69.56 ± 6.94/63.48 ± 11.76 | 66.52 ± 10.13 |
| | | | MCC | 0.342/0.507 | 0.684/0.222 | 0.422/0.282 | 0.335/0.501 | 0.282/−0.118 | 0.413 ± 0.143/0.279 ± 0.229 | 0.346 ± 0.202 |

**Table 1.** *Cont.*

| Layers | Model | Units | Indicators | The Number of Trials | | | | | Average (Validation/Test) | Total Average |
|---|---|---|---|---|---|---|---|---|---|---|
| | | | | 1 | 2 | 3 | 4 | 5 | | |
| 2 | LSTM | 64 | Acc. (%) | 80.56/69.44 | 77.78/58.33 | 83.33/61.11 | 69.44/66.67 | 61.11/77.78 | 74.44 ± 8.13/66.67 ± 6.80 | 70.56 ± 8.44 |
| | | | F1. (%) | 80.54/69.42 | 77.14/58.30 | 83.33/57.86 | 69.42/66.56 | 60.99/77.78 | 74.29 ± 8.12/65.98 ± 7.43 | 70.14 ± 8.82 |
| | | | MCC | 0.612/0.390 | 0.589/0.167 | 0.667/0.267 | 0.390/0.335 | 0.224/0.556 | 0.496 ± 0.165/0.343 ± 0.130 | 0.420 ± 0.167 |
| | | 128 | Acc. (%) | 75.00/75.00 | 75.00/58.33 | 75.00/66.67 | 69.44/66.67 | 55.56/77.78 | 70.00 ± 7.54/68.89 ± 6.89 | 69.44 ± 7.24 |
| | | | F1. (%) | 74.98/74.98 | 74.02/58.30 | 74.02/62.50 | 69.42/65.71 | 55.42/77.78 | 69.57 ± 7.34/67.85 ± 7.40 | 68.71 ± 7.42 |
| | | | MCC | 0.501/0.501 | 0.543/0.167 | 0.543/0.447 | 0.390/0.354 | 0.112/0.556 | 0.418 ± 0.163/0.405 ± 0.136 | 0.411 ± 0.150 |
| | | 256 | Acc. (%) | 63.89/75.00 | 75.00/55.56 | 69.44/61.11 | 63.89/72.22 | 52.78/83.33 | 65.00 ± 7.37/69.44 ± 9.94 | 67.22 ± 9.02 |
| | | | F1. (%) | 62.47/74.83 | 74.02/55.42 | 68.24/59.09 | 63.64/72.14 | 49.63/83.28 | 63.60 ± 8.08/68.95 ± 10.30 | 66.27 ± 9.64 |
| | | | MCC | 0.302/0.507 | 0.543/0.112 | 0.422/0.248 | 0.282/0.447 | 0.064/0.671 | 0.322 ± 0.160/0.397 ± 0.197 | 0.360 ± 0.183 |
| | GRU | 64 | Acc. (%) | 75.00/77.78 | 77.78/63.89 | 63.89/55.56 | 66.67/61.11 | 61.11/69.44 | 68.89 ± 6.43/65.56 ± 7.58 | 67.22 ± 7.22 |
| | | | F1. (%) | 74.98/77.71 | 77.14/63.64 | 61.48/53.25 | 66.56/61.11 | 61.11/69.23 | 68.26 ± 6.69/64.99 ± 8.18 | 66.62 ± 7.65 |
| | | | MCC | 0.501/0.559 | 0.589/0.282 | 0.321/0.124 | 0.335/0.222 | 0.222/0.394 | 0.394 ± 0.133/0.316 ± 0.150 | 0.355 ± 0.147 |
| | | 128 | Acc. (%) | 66.67/77.78 | 80.56/61.11 | 80.56/66.67 | 69.44/69.44 | 55.56/55.56 | 70.56 ± 9.40/66.11 ± 7.54 | 68.33 ± 8.80 |
| | | | F1. (%) | 66.56/77.78 | 80.17/61.11 | 80.54/62.50 | 69.42/68.84 | 55.42/54.29 | 70.42 ± 9.36/64.90 ± 7.93 | 67.66 ± 9.10 |
| | | | MCC | 0.335/0.556 | 0.636/0.222 | 0.612/0.447 | 0.390/0.405 | 0.112/0.118 | 0.417 ± 0.193/0.350 ± 0.158 | 0.383 ± 0.180 |
| | | 256 | Acc. (%) | 69.44/77.78 | 83.33/55.56 | 77.78/66.67 | 63.89/72.22 | 58.33/77.78 | 70.56 ± 9.06/70.00 ± 8.31 | 70.28 ± 8.70 |
| | | | F1. (%) | 69.42/77.71 | 83.13/55.00 | 77.71/62.50 | 63.64/72.14 | 58.30/77.71 | 70.44 ± 9.04/69.01 ± 8.94 | 69.72 ± 9.02 |
| | | | MCC | 0.390/0.559 | 0.684/0.114 | 0.559/0.447 | 0.282/0.447 | 0.167/0.559 | 0.416 ± 0.186/0.425 ± 0.163 | 0.421 ± 0.175 |
| | RNN-SH (tanh) | 64 | Acc. (%) | 61.11/52.78 | 83.33/61.11 | 72.22/61.11 | 61.11/66.67 | 66.67/38.89 | 68.89 ± 8.31/56.11 ± 9.69 | 62.50 ± 11.06 |
| | | | F1. (%) | 54.18/39.23 | 83.13/60.00 | 70.78/60.00 | 56.25/63.88 | 66.56/38.13 | 66.18 ± 10.50/52.25 ± 11.18 | 59.21 ± 12.89 |
| | | | MCC | 0.354/0.169 | 0.684/0.236 | 0.496/0.236 | 0.298/0.401 | 0.335/-0.228 | 0.433 ± 0.142/0.163 ± 0.210 | 0.298 ± 0.225 |
| | | 128 | Acc. (%) | 72.22/69.44 | 75.00/63.89 | 72.22/58.33 | 63.89/66.67 | 61.11/55.56 | 68.89 ± 5.39/62.78 ± 5.15 | 65.83 ± 6.09 |
| | | | F1. (%) | 70.78/69.23 | 74.02/61.48 | 71.43/55.56 | 63.64/66.56 | 54.18/44.62 | 66.81 ± 7.19/59.49 ± 8.78 | 63.15 ± 8.82 |
| | | | MCC | 0.496/0.394 | 0.543/0.321 | 0.471/0.193 | 0.282/0.335 | 0.354/0.243 | 0.429 ± 0.097/0.297 ± 0.071 | 0.363 ± 0.108 |
| | | 256 | Acc. (%) | 75.00/75.00 | 80.56/63.89 | 69.44/75.00 | 58.33/58.33 | 58.33/61.11 | 68.33 ± 8.89/66.67 ± 7.03 | 67.50 ± 8.06 |
| | | | F1. (%) | 74.98/74.51 | 80.42/63.18 | 68.24/74.98 | 57.51/56.70 | 54.04/56.25 | 67.04 ± 10.03/65.12 ± 8.23 | 66.08 ± 9.23 |
| | | | MCC | 0.501/0.521 | 0.620/0.289 | 0.422/0.501 | 0.174/0.181 | 0.211/0.298 | 0.385 ± 0.170/0.358 ± 0.132 | 0.372 ± 0.153 |
| | RNN-SH (relu) | 64 | Acc. (%) | 72.22/77.78 | 69.44/63.89 | 63.89/72.22 | 58.33/63.89 | 52.78/55.56 | 63.33 ± 7.11/66.67 ± 7.66 | 65.00 ± 7.58 |
| | | | F1. (%) | 72.14/77.71 | 66.30/60.17 | 61.48/72.22 | 57.51/63.18 | 51.85/53.25 | 61.86 ± 6.99/65.31 ± 8.69 | 63.58 ± 8.08 |
| | | | MCC | 0.447/0.559 | 0.491/0.351 | 0.321/0.444 | 0.174/0.289 | 0.058/0.124 | 0.298 ± 0.163/0.354 ± 0.146 | 0.326 ± 0.158 |
| | | 128 | Acc. (%) | 69.44/75.00 | 80.56/58.33 | 63.89/63.89 | 63.89/63.89 | 61.11/66.67 | 67.78 ± 6.94/65.56 ± 5.44 | 66.67 ± 6.34 |
| | | | F1. (%) | 67.41/74.51 | 80.54/55.56 | 61.48/63.86 | 63.64/63.18 | 60.00/62.50 | 66.61 ± 7.40/63.92 ± 6.08 | 65.27 ± 6.90 |
| | | | MCC | 0.449/0.521 | 0.612/0.193 | 0.321/0.278 | 0.282/0.289 | 0.236/0.447 | 0.380 ± 0.136/0.346 ± 0.120 | 0.363 ± 0.129 |
| | | 256 | Acc. (%) | 66.67/72.22 | 77.78/63.89 | 61.11/63.89 | 58.33/55.56 | 52.78/50.00 | 63.33 ± 8.50/61.11 ± 7.66 | 62.22 ± 8.16 |
| | | | F1. (%) | 66.25/71.88 | 77.71/62.47 | 57.86/63.64 | 57.51/53.25 | 42.86/37.69 | 60.44 ± 11.46/57.78 ± 11.65 | 59.11 ± 11.63 |
| | | | MCC | 0.342/0.456 | 0.559/0.302 | 0.267/0.282 | 0.174/0.124 | 0.101/0.000 | 0.288 ± 0.158/0.233 ± 0.157 | 0.261 ± 0.160 |

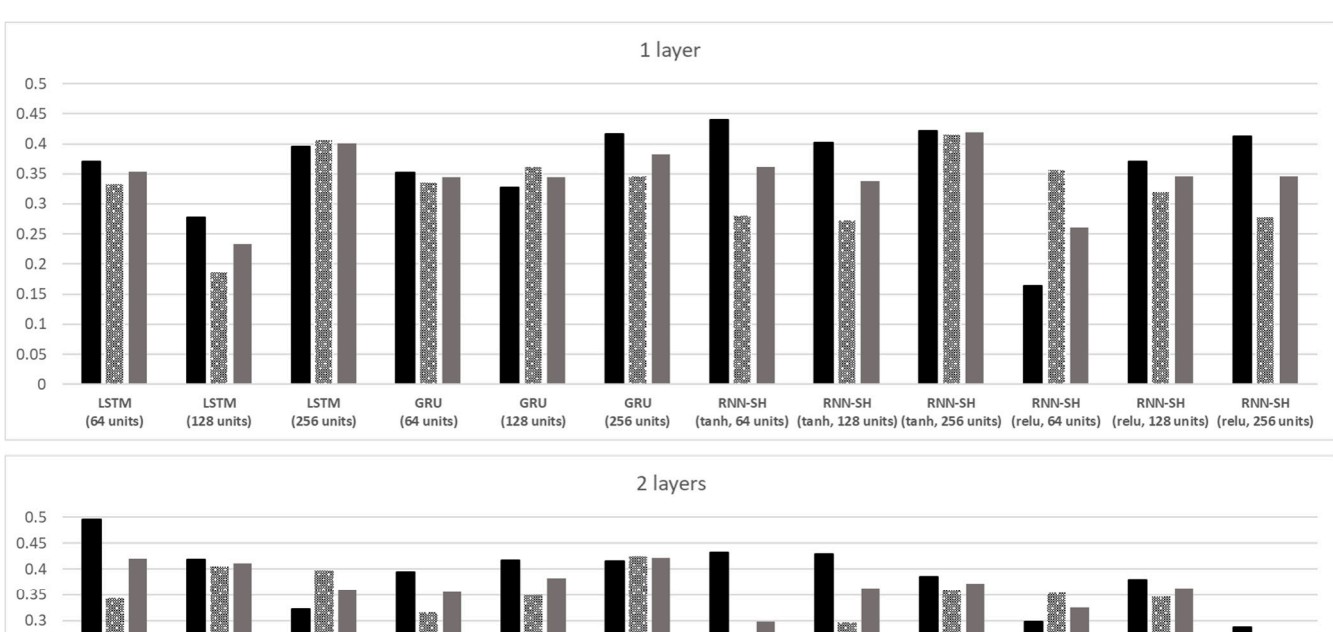

**Figure 7.** The graphs of the average value of each model in Table 1. The left is the validation average, the middle is the test average, and the right is the total average. Here, the standard deviation is omitted to make the graphs easier to see.

As a result, from Table 1, there was no significant difference in the performance between RNNs. Statistically significant difference was not found between the top three models (GRU: two layers and 256 units, LSTM: two layers, 64 units, and RNN-SH (tanh): one layer and 256 units) by the Friedman test. Here, Figure 8 shows the average ROC curve with AUC values of RNN-SH and GRU, and Figure 9 shows a graph of the relationship between the number of units and the parameters of the PD detection model. From Figure 8, it is clear that the AUC value of GRU is higher than the value of RNN-SH (tanh); however, from Figure 9, the parameters of RNN-SH (tanh) with one layer and 256 units are less than a third of the parameters of GRU with two layers and 256 units, although the difference in the performance is small. In addition, the parameter amount of LSTM with two layers and 64 units is not much different from that of RNN-SH with one layer and 256 units. However, in the case of the LSTM, though the total average is high, there is a difference between the validation average and the test average, and it can be confirmed that overfitting has occurred. On the other hand, for the RNN-SH, the validation average and test average are about the same value. Figure 10 shows the examples of confusion matrix in the first trial of LSTM and RNN-SH (tanh). In Figure 10, LSTM has a better performance for the validation set than RNN-SH, but not so much for the test set.

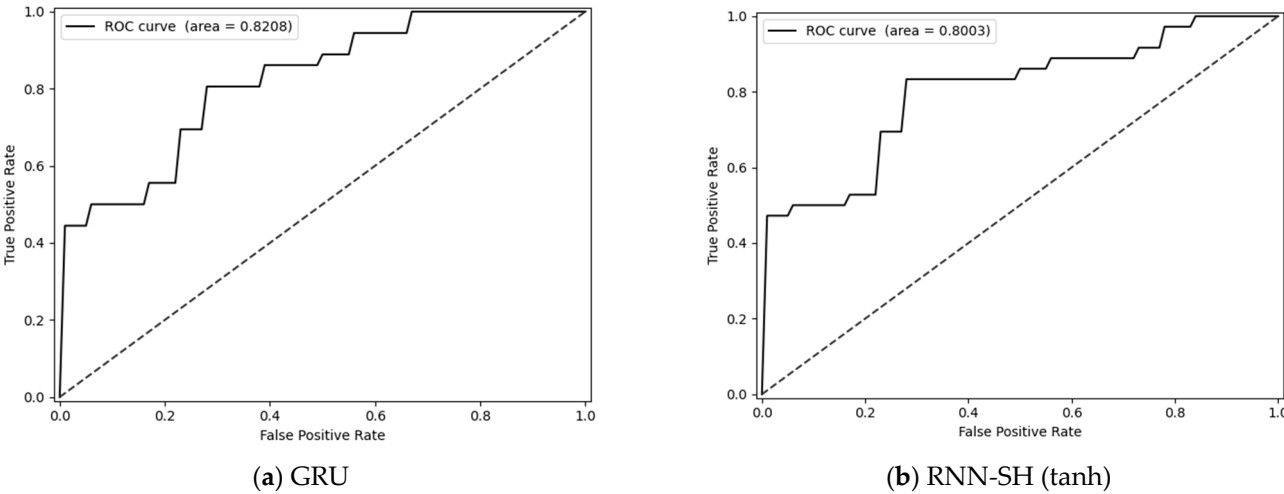

**Figure 8.** The average ROC curve with AUC values of GRU and RNN-SH. (**a**) is the ROC curve of the GRU, and (**b**) is that of the RNN-SH (tanh). The average here is an average of the validation set and the test set.

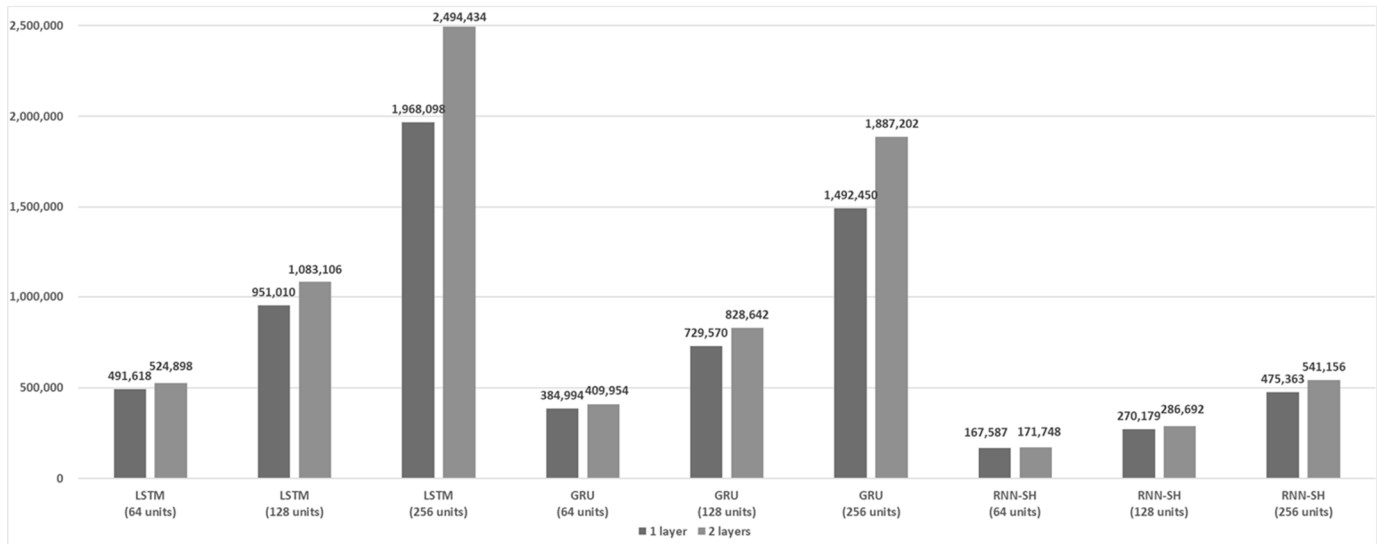

**Figure 9.** The graph of the relationship between the number of units and the number of parameters of PD detection model. If RNNs have the same number of units, the number of parameters of PD detection model using RNN-SH is considerably reduced.

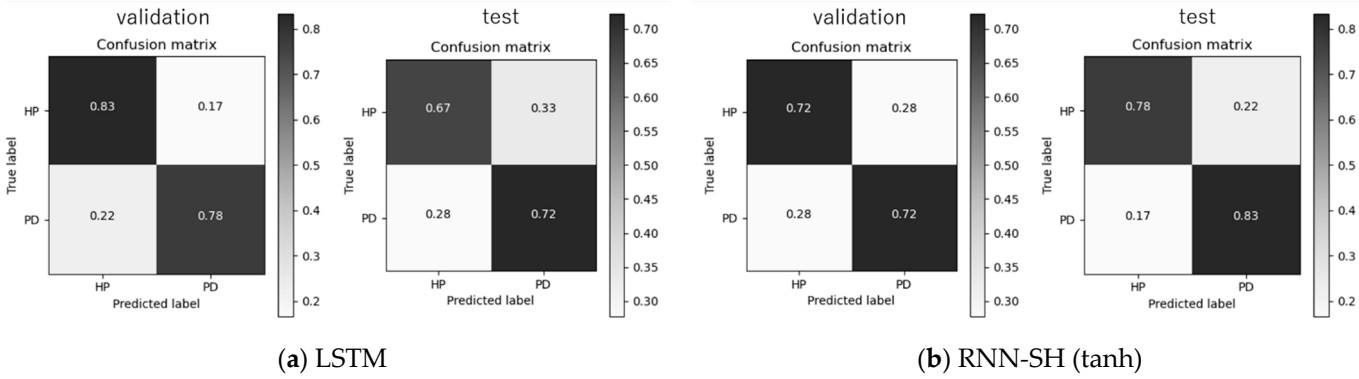

**(a)** LSTM     **(b)** RNN-SH (tanh)

**Figure 10.** The examples of confusion matrix in first trial of LSTM and RNN-SH (tanh). (**a**) is the confusion matrix of the LSTM, and (**b**) is that of the RNN-SH (tanh). Here, the confusion matrix is normalized.

From Table 1 and Figure 7, in RNN-SH, the performance when the output activation function was tanh was higher than that of relu.

Figure 11 shows the execution time of each RNN in the case of one layer and 256 units in the first execution. Here, since we used LSTM and GRU that were pre-implemented in pytorch, accurate speed comparison with RNN-SH, which was implemented by the combination of pytorch functions, cannot be performed. Therefore, in Figure 11, we used LSTM and GRU that were individually implemented by the combination of pytorch functions, and the speeds are compared. In order to perform speed comparisons fairly, we considered the parallelism of LSTM and GRU, and implemented LSTM and GRU so that they could be calculated at high speed by splitting the result of parallel calculation for each weight of gate, etc. From Figure 11, since RNN-SH has fewer parameters and a lower calculation cost than LSTM and GRU, the learning of RNN-SH was faster than them.

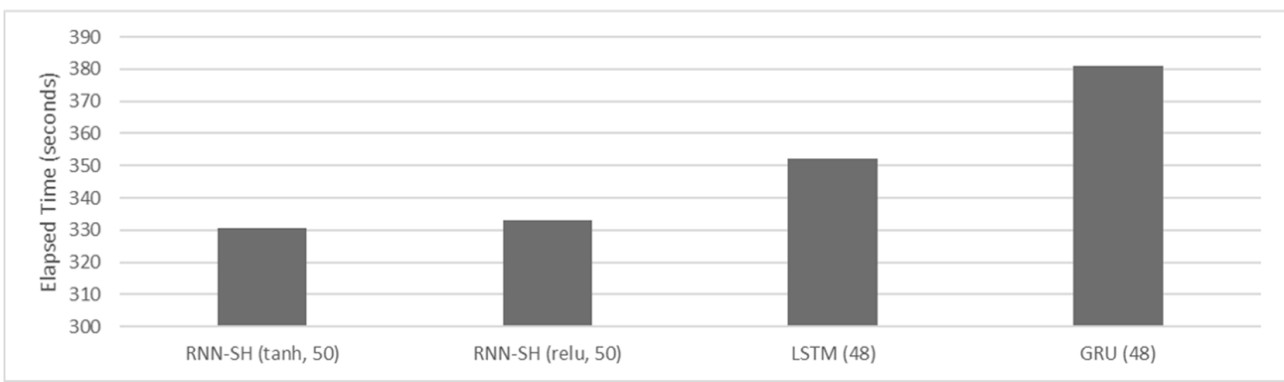

**Figure 11.** The execution time of each RNN in the case of 1 layer and 256 units. The number in parentheses is the epoch at the time when the loss was the lowest for the validation set. RNN-SH is faster than LSTM and GRU. GRU is slow due to its low parallelism.

Figure 12 shows learning curves of loss for each RNN with 256 units and one or two layers. From Figure 12, it can be seen that the losses oscillate due to the small amount of dataset, but the learning of RNN-SH progresses relatively gently compared to that of LSTM and GRU. However, RNN-SH (tanh) with two layers oscillate violently and could not learn well.

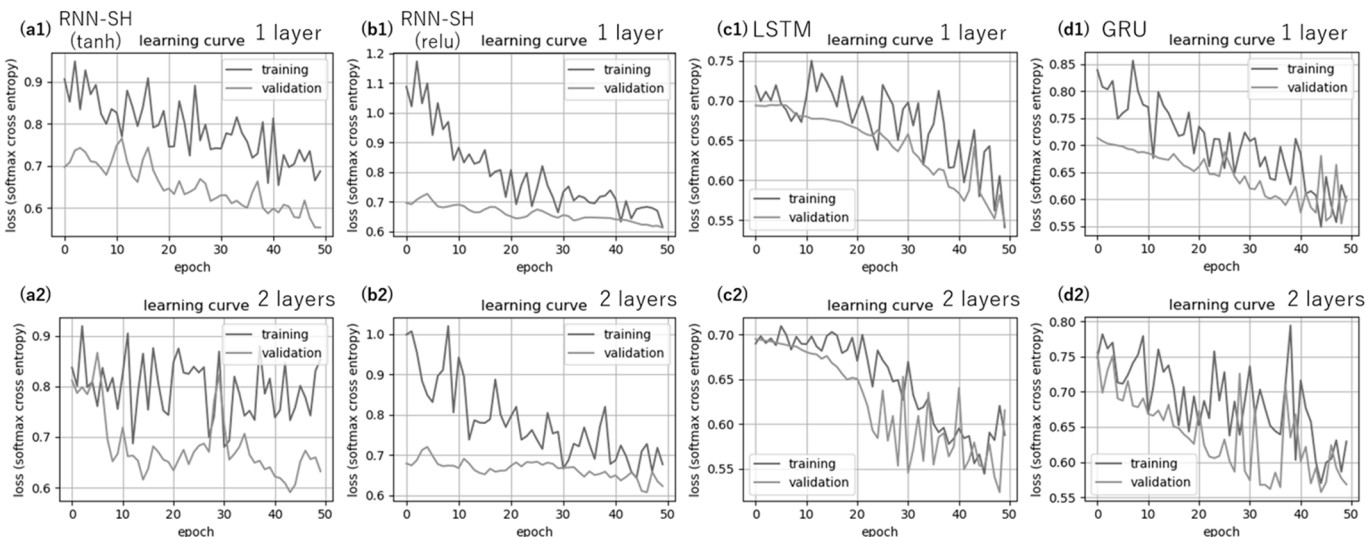

**Figure 12.** Learning curves of loss for each RNN with 256 units and one or two layers. (**a1**,**a2**), (**b1**,**b2**), (**c1**,**c2**), and (**d1**,**d2**) represent the graphs of learning curves of RNN-SH (tanh), RNN-SH (relu), LSTM, and GRU, respectively. The number after the letters corresponds to the number of layers.

### 4.5. Discussion

Statistically significant difference was not found between RNN models, but the total number of parameters of the PD detection model using RNN-SH is about 1/4 of the model using LSTM and about 1/3 of the model using GRU when the number of units and layers are the same. This difference becomes even wider when the units increase or multiple layers are used. Hence, it is very effective in terms of memory efficiency and faster learning. When using RNN-SH, it is easy to take measures against overfitting because there are few parameters, slow learning progresses, and only weight parameters for the input.

Regarding the comparison when tanh and relu are used in RNN-SH, taking the case of one layer and 256 units for example, the scalar value *a*, which is the gate parameter, is about 1.007 for tanh and about 1.012 for relu, and they are very close with each other. This was also the case under other conditions in these experiments. Therefore, it is considered that the difference in the performance of tanh and relu is caused by the difference only in activation function. From Figure 12, RNN-SH (tanh) with 256 units and two layers oscillate violently, and the reason why it could not learn well comes from the vanishing gradient at the output due to tanh. On the other hand, RNN-SH (relu) with 256 units and two layers could be learned smoothly; however, the accuracy was lower than that of tanh. Here, Figure 13 shows the examples of confusion matrix in the first trial of the tanh and relu RNN-SH. From Figure 13, the FP rate of relu was higher than that of tanh, and TN rate of relu was lower than that of tanh, indicating that the relu model could not identify PD patients properly and could not be learned well. Relu has gathered attention as the activation function that contributes to the multi-layering of neural networks. However, relu is not symmetric with respect to the origin and has problems such as a dying relu problem [30]. In our experiments, we did not use optimal parameter initialization or normalization methods that optimize for relu. Thus, dying relu may have occurred and caused the accuracy to deteriorate. Since relu is the function that is important for multi-layering, it is necessary to analyze in the future what is needed to improve the performance when using the relu.

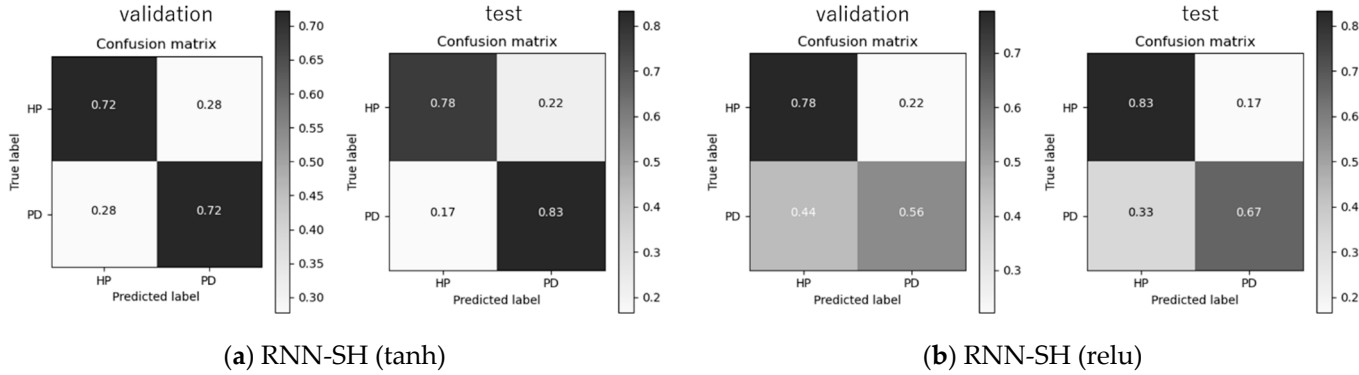

(**a**) RNN-SH (tanh)          (**b**) RNN-SH (relu)

**Figure 13.** The examples of confusion matrix in first trial of RNN-SH. (**a**) is the confusion matrix of the RNN-SH (tanh), and (**b**) is that of the RNN-SH (relu). In first trial case, although the model could be learned relatively well when tanh is used, it could not be learned well when relu is used, and the PD detection ability was low.

In terms of PD detection using an RP, the accuracy is around 70%. This is because some of the voice data of healthy people had hoarse voices and uneven voice volume. In our experiments, the voice/a/ was used; however, it was difficult for even a healthy person to utter the voice/a/ at a constant volume for a long time. Therefore, from the results of our experiments, the voice/a/ was not sufficient for PD detection. In order to improve the accuracy of PD detection, it is necessary to consider what pronunciation is more suitable for the detection in the future. Additionally, in our experiment, the RP images were resized by bilinear interpolation in consideration of the memory usage. However, when the RP image is compressed, features different from the original features may appear or important features may disappear. Since it is possible that the accuracy had deteriorated due to resizing, it is necessary to further study the image resizing method and the architecture of the CNN in the future.

## 5. Conclusions

This paper evaluated the effectiveness of our RNN-SH model in a practical medical application of PD detection using RP with less parameters than other gated RNNs such as LSTM and GRU. RNN-SH can greatly reduce parameters, more so than other RNNs, with comparable accuracy for time series data processing although the difference in accuracy was small and a statistically significant difference was not found between RNN models.

Since the gate parameter of RNN-SH is scalar, it is easy to analyze the result. In addition, another advantage is that the activation function can be easily changed according to the tasks. However, it turned out that it is difficult to improve the accuracy by simply replacing tanh with relu as the activation function in RNN-SH. From our experiment, the PD detection ability is lower using relu in comparison to using tanh, and we consider that dying relu might be the cause. It is necessary to analyze and make further improvements by considering parameter initialization and normalization methods, etc.

Since the dataset size in our experiment was relatively small, the proposed method will see more improvement by increasing the data in the future. More analysis on the input sound type, the RP image size, and the deep learning structures will be included in our future work for further improving the performance of PD detection from voice. In regard to RP image compression, we would investigate an image size or a CNN structure that does not deteriorate the accuracy of PD detection in more detail to improve the current model.

**Author Contributions:** Conceived and designed the model, T.F. and Z.L.; performed the experiment and analyzed the results, T.F.; wrote the preliminary version of this manuscript, T.F.; revised the manuscript, Z.L., C.Q., K.M., and S.C. All authors have read and agreed to the published version of the manuscript.

**Funding:** This research received no external funding.

**Institutional Review Board Statement:** All procedures performed in studies involving human participants were in accordance with the ethical standards of the institutional and/or national research committee and the "Law of the People's Republic of China on Medical Practitioners" (1998) declaration and its later amendments or comparable ethical standards.

**Informed Consent Statement:** Informed consent was obtained from all subjects involved in the study.

**Data Availability Statement:** Due to the nature of this research, participants of this study did not agree for their data to be shared publicly at present, so supporting data is not available.

**Acknowledgments:** We wish to express our appreciation to the editor and reviewers for their insightful comments as well as the participants for supporting this research.

**Conflicts of Interest:** The authors declare no conflict of interest.

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
