# Peer review of "Performance Evaluation of RNN with Hyperbolic Secant in Gate Structure through Application of Parkinson’s Disease Detection"

_applsci, doi:10.3390/app11104361_

Round 1
Reviewer 1 Report
- The work is interesting because of the practical application in the medical domain, as it could help in the diagnosis of Parkinson's disease. However,
at the research level there is not a great contribution in this paper, since it uses a RNN model that the authors had already proposed in another previous work and here they simply show an example of practical application.
- Since the main motivation of the paper is its contribution to the detection of Parkinson's disease, the state of the art does not seem to be the most
suitable, because only one reference appears about this topic.
- The explanation of what steps they take to adapt the diagnostic aid problem using their RNN model and converting the problem into a time series problem is interesting.
- The authors describe the dataset used, how they build it, how balanced data, etc., but after, they do not analysis in deep de results according the data. I mean that maybe the data used must be improved, for example, increasing it because it could be small for learning the RNN model, etc. There is no information about that, and also nothing about the data others authors used for parkinson detection.
- The paper showed the that their RNN-SH model can be used in PD detection with less parameters than other gated RNNs such as LSTM and GRU, but the improvement is not significant. And most important, regarding PD detection there is a great room of improvement. Therefore, What is the real contribution of the paper? Because the conclusions on their proposed NRN model will already be in the other paper they are going to publish. In this sense, the contribution of this work is weak.
Reviewer 2 Report
- Question: What is difference between average column and all column in the table 1. Describe the differences between them or each meaning.
- Draw the performance value in a figure, in which values of the table 1 reflect. A reader can easily notice the performance of each model with the figure.
- I thank any performance of each model you experimented are same in your table 1 because all average values are in the same confidence level (even with 95%). Can you test your model’s mean value is different from the others ? . If the null hypothesis is rejected, your model is superior, otherwise, the performance is same in the view of performance.
Reviewer 3 Report
The paper proposes a Recurrent Neural Network (RNN) architecture with sech function in the gate for Parkinson's disease (PD) detection from the voice (speech) data.
Comments:
- Add a few sentences to the abstract summarizing the main experimental results,
- Present an outline of the structural organization of the paper at the end of the Introduction section.
- The overview of related works is very limited and does not include many important works published recently. The discussion could be organized into machine learning approaches and deep learning approaches and “X-Vectors: New Quantitative Biomarkers for Early Parkinson's Disease Detection From Speech”, “Detecting parkinson's disease with sustained phonation and speech signals using machine learning techniques”, and “Detection of speech impairments using cepstrum, auditory spectrogram and wavelet time scattering domain features” could be discussed among other works. Summarize the limitations of the related works as a motivation of your study.
- How do you deal with the overfitting problem using the proposed network architecture?
- Line 146. Subsection numeration is wrong.
- Line 190. Why 35th percentile is used?
- What is the age and gender distribution of subjects in the dataset, and the severity level of the Parkinson’s patients?
- Is the voice dataset available openly? If not, make it public and share it with the research community.
- Table 1: “The number of executions” – it is misleading. Consider revising.
- Present and discuss the classification confusion matrices.
- Present and discuss the ROC curves with the AUC values.
- The difference between methods’ performance is small. Is it statistically significant? The results should be analyzed statistically. Use the non-parametric Friedman test and the post-hoc Nemeny test to compare methods. Represent mean ranks of methods visually using the Critical Distance (CD) diagrams.
- There are some typos (see “?−?????”). Carefully check the language.
- The conclusions are weak. Present more insights into the advantages/disadvantages of different architectures and activation functions. Use the results of the experiments and statistical analysis to support your claims.
Round 2
Reviewer 1 Report
The authors have introduced several changes in the paper, making it more consistent and better understood. They have also improved the analysis of results and future work, which helps to better understand the extent of their current proposal.
Reviewer 3 Report
The revisions improved the quality of the paper. I have only a few more minor comments:
- Sech is not a commonly used abbreviation. I suggest to change it to “hyperbolic-secant activation function”, or similar.
- Figure 6: add a label to the y-axis of the plots.
- Line 265: “??,??,??,?? are true positives, …” should be “??,??,??,?? are the number of true positives, …”
- Line 268: present a motivation of using MCC.
- The caption of figure 9: “the parameters of PD” should be “the number of parameters of PD”.
